# In Vitro Measurement and Mathematical Modeling of Thermally-Induced Injury in Pancreatic Cancer Cells

**DOI:** 10.3390/cancers15030655

**Published:** 2023-01-21

**Authors:** Faraz Chamani, Marla M. Pyle, Tej B. Shrestha, Jan Sebek, Stefan H. Bossmann, Matthew T. Basel, Rahul A. Sheth, Punit Prakash

**Affiliations:** 1Department of Electrical and Computer Engineering, Kansas State University, Manhattan, KS 66506, USA; 2Department of Anatomy and Physiology, College of Veterinary Medicine, Kansas State University, Manhattan, KS 66506, USA; 3Nanotechnology Innovation Center of Kansas State (NICKS), Kansas State University, Manhattan, KS 66506, USA; 4Department of Cancer Biology, University of Kansas Medical Center, Kansas City, KS 66160, USA; 5Department of Interventional Radiology, University of Texas MD Anderson Cancer Center, Houston, TX 77030, USA

**Keywords:** hyperthermia, pancreatic cancer, cell death, Arrhenius injury model, thermal damage

## Abstract

**Simple Summary:**

Thermal therapies, the controlled heating of tissue, are a clinically accepted modality for the treatment of localized cancers and are under investigation as part of treatment strategies for pancreatic cancer. The bioeffects of heating varies as a function of intensity and duration of heating and can vary across tissue types. We report on the measurement of thermal injury parameters for pancreatic cancer cell lines in vitro and assess their suitability for predicting changes in cell viability following heating. The results of this study may contribute to research investigating the use of thermal therapies as part of pancreatic cancer treatment strategies, the development of modeling tools for predictive treatment planning of thermal therapies, and understanding the effects of other energy-based interventions that may involve perturbation of tissue temperature.

**Abstract:**

Thermal therapies are under investigation as part of multi-modality strategies for the treatment of pancreatic cancer. In the present study, we determined the kinetics of thermal injury to pancreatic cancer cells in vitro and evaluated predictive models for thermal injury. Cell viability was measured in two murine pancreatic cancer cell lines (KPC, Pan02) and a normal fibroblast (STO) cell line following in vitro heating in the range 42.5–50 °C for 3–60 min. Based on measured viability data, the kinetic parameters of thermal injury were used to predict the extent of heat-induced damage. Of the three thermal injury models considered in this study, the Arrhenius model with time delay provided the most accurate prediction (root mean square error = 8.48%) for all cell lines. Pan02 and STO cells were the most resistant and susceptible to hyperthermia treatments, respectively. The presented data may contribute to studies investigating the use of thermal therapies as part of pancreatic cancer treatment strategies and inform the design of treatment planning strategies.

## 1. Introduction

Pancreatic cancer is the fourth leading cause of cancer-related death in the United States and accounts for 8% of cancer deaths, with a low five-year survival rate of approximately 10% [1,2]. Surgical resection remains the most effective treatment strategy; however, only approximately 20% of patients are surgical candidates at the time of diagnosis [2,3]. For patients with unresectable pancreatic cancer, the use of chemotherapy alone or in conjunction with surgery remains the gold standard, although long-term survival rates are poor and the regimen comes with risks for major complications in patients with advanced disease [4,5]. Thermal ablation [6,7] and other non-ionizing energy-based local interventions such as irreversible electroporation are under investigation as potential adjuvant or stand-alone treatment options for patients with unresectable pancreatic adenocarcinoma [8]. In addition to ablative effects, heating in the mild hyperthermia range (39–43 °C) may offer a means for thermally-triggered drug delivery [9,10,11] or serve as an adjuvant to ionizing radiation and/or chemotherapy [12,13,14,15].

The bioeffects induced by heating are a function of the time-temperature profile during heating and may vary across cell types. Mathematical models relating changes in cell viability, stress protein expression, and other biomarkers to the time-temperature history during heating have been reported [16]. Cell/tissue-specific parameters for these models can be determined from experiments on cells in vitro [17,18,19]. One of the most widely used models is the Arrhenius thermal injury model [18], which describes cell death following heating as a first-order exponential relationship between temperature and duration of heating, and has been applied to assess thermal damage in various cell types, including liver cancer cells [20], prostate tumor cells [21], and breast cancer cells [22]. The thermal isoeffective dose model, which relates an arbitrary transient temperature profile to equivalent minutes of heating at a reference temperature, typically taken to be 43 °C, is derived from the Arrhenius model [23]. Despite its wide usage, the standard Arrhenius model fails to represent thermally-induced injury or cell death at mild hyperthermic temperatures (39–43 °C) for several cell types, showing significant over-prediction of the initial “shoulder” region as explained by Pearce [18]. Augmenting the Arrhenius model with a time delay term has been proposed to account for the delayed cell death at low temperatures [24]. Other models for thermal injury have been proposed, including a two-state statistical thermodynamic model by Feng et al. [19] and a three-compartment reaction-based model by O’Neill et al. [25]. While thermal injury parameters for a range of cell types have been reported, there are few published data reporting on the viability of pancreatic cancer cells following heating. Identification of thermal injury parameters is important to inform the design of thermal therapy devices and systems, select treatment doses, and to inform interpretation of experimental and clinical studies involving heat as a therapeutic modality.

The objective of the present study was to determine the kinetics of thermal injury to pancreatic cancer cells in vitro following thermal exposure to temperatures up to 50 °C and use these data to evaluate predictive models for thermal injury. Given the central role of experimental murine models in pancreatic cancer research, we conducted studies on two murine pancreatic cancer cell lines (KPC and Pan02), as well as a normal murine fibroblast cell line (STO). The KPC murine model (KRAS/TP53 point mutation) [26] is an established genetically-engineered and clinically relevant model of pancreatic ductal adenocarcinoma that represents many histopathological features observed in human disease. The murine pancreatic adenocarcinoma cell line Pan02 [27,28], syngeneic to C57BL/6, is an established grade III model widely used for pre-clinical evaluation of single and combination therapies. Given the significance of the stroma in pancreatic tumors, we also evaluated the kinetics of thermal injury on STO cells. For each of these cell lines, monolayer cell cultures were heated in water baths to temperatures in the range 42.5–50 °C for 3–60 min, and cell viability following heating was assessed up to 24 h following hyperthermia and compared to 37 °C control. The kinetics of thermal injury were estimated from the measured viability data. Finally, we comparatively assessed three mathematical models for predicting thermally-induced changes in cell viability based on the measured in vitro data.

## 2. Materials and Methods

### 2.1. Cell Culture

KPC and STO cells were cultured with DMEM’s medium (Gibco™ 11995065, Fisher Scientific, Hampton, NH, USA) with 10% fetal bovine serum (Corning™ 35015CV, Fisher Scientific) and 1% penicillin-streptomycin (Gibco™ 15140148). Pan02 cells were cultured with RPMI 1640 Medium (Gibco™ 11875093, Fisher Scientific) supplemented with sodium pyruvate and 10% fetal bovine serum. Cultures were maintained in a 37 °C, 5% CO_2_ incubator in 75 cm^2^ phenolic culture flasks. In preparation for hyperthermia treatment, cells were seeded in *n* = 6 wells of 96-well culture plates at a density of ~30,000 cells/cm^2^ at a volume of 200 μL medium/well and maintained in a 37 °C, 5% CO_2_ incubator for 24 h to allow cells to reach log phase prior to hyperthermia.

### 2.2. In Vitro Hyperthermia to Monolayer Cell Cultures

To expose cells in culture to hyperthermia, sealed 96-well plates containing cells were immersed in temperature-controlled water baths (shown to be an effective method for accurate and uniform heating of cell culture samples) [29]. To assess temperatures during hyperthermia, transient temperature profiles were recorded using five T-type thermocouples embedded within distinct wells of a dummy plate that contained no cells while filled with 200 μL of water/well. The dummy well plate was immersed in water baths simultaneously with the cell-containing plate, thus providing a reasonable assessment of the temperatures within the cell-containing wells. Figure 1 illustrates the dummy plate design, including five thermocouples positioned and sealed within four corner wells and one central well.

Since the time to reach the setpoint temperature can be rather slow, we employed a two-step approach. First, both the cell-containing and dummy well plates were immersed in a water bath set at an elevated temperature of ~80 °C. When the temperature recorded by thermocouples in the dummy plate reached within 0.2 °C of the target temperature (i.e., 42.5, 44, 46, or 50 °C), plates were immediately transferred to another pre-heated water bath that was set to the desired target hyperthermic temperature for a predetermined duration in the range of 3–60 min. A USB thermocouple data acquisition module (TC-08 OMEGA) was used to record the temperature data from the thermocouples embedded within the dummy plate. Following hyperthermia treatment, sealing films were removed and the 96-well culture plates were returned to a 37 °C incubator for subsequent 6 h and 24 h recovery of thermal injury. For each cell line, an additional plate containing cells was also immersed in a 37 °C water bath for the experimental durations considered in this study, providing a no-hyperthermia control.

### 2.3. Cell Viability Evaluation

After an incubation period of 6 h and 24 h post-heating (shown to be effective evaluation periods for measuring cell viability [24]), the cell culture supernatant was discarded from each 96-well culture plate and viability was determined using the 3-(4,5-dimethylthiazol-2-yl)-2,5-diphenyl-2H-tetrazolium bromide (MTT) colorimetric assay [30], which is based on the reduction of a yellow tetrazolium salt to purple formazan crystals by metabolically active cells. The measured optical density for each time-temperature combination was normalized to the optical density measured for no-heat control plates immersed in a 37 °C water bath for the same time duration. The normalized values thus represent the average concentration of viable cells across *n* = 6 wells following hyperthermia exposure for each experimental group.

### 2.4. Thermal Injury Analysis

#### 2.4.1. Arrhenius Model of Thermal Injury

The Arrhenius cell injury method models cell death as a first order chemical reaction where the source materials (viable cells) are transformed to the product (non-viable cells). After identification of the rate parameters for the reaction, the Arrhenius model allows prediction of cell injury for arbitrary time-temperature profiles. Equations (1) and (2) describe the Arrhenius model:(1)$$\Omega \left(t\right)=\ln\left(\frac{C_{0}}{C\left(t\right)}\right)$$
(2)$$\Omega \left(t\right)=A\int_{0}^{t}e^{\frac{-E_{a}}{RT\left(\tau \right)}}d\tau ,P=100(1-e^{-\Omega })$$
where *C*_0_ is the initial concentration of live cells prior to thermal exposure, *C*(*t*) is the concentration of live cells after *t* seconds of heating, *Ω*(*t*) is a positive number representing the extent of thermal damage at time *t*, *A* is the frequency factor (s^−1^), *E_a_* is the activation energy (J/mole), *R* is the universal gas constant (8.31 J⋅K^−1^⋅mol^−1^), and *T* is temperature [K]. The value of *Ω*(*t*) can be cast as a probability, *P*, of thermally-induced injury. The parameters of the model, *A* and *E_a_*, are cell line specific and can be determined from experiments where cells are exposed to isothermal heating. As the first step, the rate of decay in cell viability (*k*) can be determined from viability measurements following heating as a function of time at multiple temperatures [16] by fitting Equation (3) to the experimentally measured cell survival, *S*.
(3)$$S=e^{-kt}$$
(4)$$\ln\left(k\right)=\ln\left(A\right)-\frac{E_{a}}{RT}$$

Then, using Equation (4), the relationship between the natural logarithm of the constant (*k*) and the reciprocal of temperature (1/*T*) is plotted to find *A* and *E_a_* from the slope and y-intercept of the fit, respectively.

#### 2.4.2. Arrhenius Thermal Injury Model with Time Delay

As described by Feng et al. [19] and Pearce et al. [24], some cell lines initially exhibit a significant shoulder region where cell viability remains high until a threshold lethal thermal dose is attained. The conventional Arrhenius model may not accurately represent changes in cell viability for these cells. To address this limitation of the standard Arrhenius thermal injury model, an improved Arrhenius model was presented by Pearce et al. [24] by adding a temperature-dependent time delay (*t_d_*) using the slope (*m*) and intercept (*b*) to compensate for the measured viability data within the shoulder region:(5)$$t_{d}=b-mT$$
(6)$$\Omega \left(t\right)=\left\{\begin{array}{c} 0,t<t_{d}\\ A\int_{0}^{t}e^{\frac{-E_{a}}{RT\left(\tau \right)d\tau }},t\geq t_{d} \end{array}\right.$$
where *t* represents total heat exposure duration, *t_d_* denotes the time delay in seconds, *T* is the temperature in Kelvin, and *m* and *b* represent relevant coefficients obtained by slope and intercept of the equation, respectively. The ordinary Arrhenius injury process is initiated when *t > t_d_* and is calculated from that point forward.

#### 2.4.3. Two-State Thermal Injury Model

Feng et al. [19] presented a two-state cell damage model under hyperthermia conditions, which was reported to be in good agreement with experimental data. In their study, a general two-state model was proposed to characterize the entire cell population with two distinct and measurable subpopulations of cells, in which each cell is in one of the two substates, either viable (live) or damaged (dead). The resulting cell viability can be expressed as follows:(7)$$C\left(\tau ,T\right)=\frac{e^{\left(\frac{-\Phi \left(\tau ,T\right)}{KT}\right)}}{1+e^{\left(\frac{-\Phi \left(\tau ,T\right)}{KT}\right)}}$$
(8)$$\ln\left(\frac{C\left(\tau ,T\right)}{1-C\left(\tau ,T\right)}\right)=\left(\frac{\gamma }{T}\right)-\beta -\alpha \tau$$
(9)$$C\left(\tau ,T\right)=\frac{e^{\left(\frac{\gamma }{T}\right)-\beta -\alpha \tau }}{1+e^{\left(\frac{\gamma }{T}\right)-\beta -\alpha \tau }}$$

Φ(τ, T*)* was defined as a function that is linear in exposure time τ when the temperature T is fixed and *K* is constant. In their study, in vitro cell viability data from hyperthermia experiments on human PC3 prostate cancer cells and normal RWPE-1 cells were compared against the two-state damage model and used to determine the parameters in the function *Φ*(τ, T). This model requires three experimentally derived fit coefficients (α, β and γ) that were estimated using a standard bilinear least-squares regression algorithm. Finally, the fractional cell survival at any time point can be calculated using Equation (9).

### 2.5. Determination of Heat-Induced Thermal Dose (CEM 43)

The Sapareto–Dewey thermal isoeffective dose model is a means to compare thermal damage accumulated after heating with an arbitrary time-temperature profile against *t*_43_, the equivalent time needed to achieve the same level of damage when heated to 43 °C (CEM 43) [31,32]. *t*_43_ can be calculated with Equation (10).
(10)$$t_{43}=\sum_{i=1}^{n}t_{i}R_{CEM}^{\left(43-T_{i}\right)}$$
where *t*_43_ is the cumulative number equivalent time (min) at 43 °C, $T_{i}$ is the temperature at the *i*-th time interval *t_i_*, and *R_CEM_* is 0.5 when *T_i_* > 43 °C and *R_CEM_* is 0.25 when *T_i_* ≤ 43 °C. In Equation (10), *R_CEM_* represents the rate at which time taken to achieve a thermal damage isoeffect drops for each unit rise in temperature.

### 2.6. Model Assessment

The accuracy of our developed injury predictive models was assessed on murine KPC pancreatic cancer cell lines that were exposed to non-isothermal heating to temperature in the range 47–51 °C. A coupled electromagnetic–bioheat transfer computational model simulating microwave thermal ablation (MWA, 50 W, 10 min with a 14 G water-cooled applicator), as described in our prior studies [33], was used to identify time-temperature profiles at the periphery of the ablation zone. Detailed information regarding the heat transfer model, model parameters, and the numerical method are provided in the Appendix A.

Finally, in vitro hyperthermia experiments were performed to expose KPC cells to temperature profiles similar to those at the periphery of the ablation zone. The measured cell viability was compared against model predictions.

## 3. Results

### 3.1. Temperature Profiles in Dummy Well Plates

Figure 2 shows the measured temperature profile inside five wells of the dummy 96-well plate during a 46 °C hyperthermia exposure. Parameters used to quantitatively assess the heating profiles are also illustrated, including: ramp time, duration of the steady-state phase, target error, homogeneity of heating, and duration of the cool-down phase. Table 1 lists the mean values and ranges for each of these parameters across heating experiments for target setpoint temperatures of 42.5, 44, 46, and 50 °C for all three cell lines considered in this study. Accuracy represents the error between the target temperature and mean recorded temperature based on five sealed thermocouples during the constant heating phase, ramp time represents the time required to reach the target temperature from physiological temperature (37 °C), and the cooling phase represents the time it takes to drop to physiological temperature from target temperature following hyperthermia exposure.

### 3.2. Cell Viability Measurement

Figure 3 shows the measured cell viability assessed using the MTT assay for all three cell lines at 6 h and 24 h post hyperthermia.

### 3.3. Arrhenius Thermal Damage Models

Figure 4a illustrates the relationship between ln (k) and 1/T for data measured at 24 h post-heating. The thermal damage kinetic coefficients A and *E_a_* are determined from the intercept and slope, respectively, of the best-fit line to the data.

Table 2 lists the thermal damage kinetic parameters of *E_a_*, *A*, and time delay parameters (*m*, *b*) that were calculated from the measured viability data 24 h post hyperthermia for each of the three cell types.

The coefficient of determination (R^2^) for calculated kinetic parameters was in the range of 95–99%, indicating the suitability of the Arrhenius injury models for predicting the extent of heat-induced cell injury. The measured and calculated damage were compared as shown in Figure 5a,b for the simple Arrhenius model and the Arrhenius model with time delay, respectively. The Root Mean Square Error (RMSE) for the simple Arrhenius model and the Arrhenius thermal damage model with time delay were 12.24% and 8.48%, respectively.

### 3.4. Two-State Model of Thermal Damage

The measured and calculated damage were compared as described in Figure 5c. Cell viability data across all considered thermal doses in all three cell lines was investigated. RMSE for 6 h and 24 h recovery was 31.66% and 51.22%, respectively.

### 3.5. CEM 43 Calculation

Table 3 lists the *R_CEM_* values measured in the present study and compares against *R_CEM_* values for other cell types reported in the literature.

Figure 6b illustrates the mean value of recorded temperature based on five sealed thermocouples in the 96-well dummy plate, as well as clinically relevant simulated temperature time history. Figure 6c shows the comparison between the measured and calculated percentage of cell survival following hyperthermia exposures that were obtained by MTT assay and our developed predictive models, respectively.

## 4. Discussion

Knowledge of thermal sensitivity of representative target cells is informative for the design and optimization of thermal therapy protocols (i.e., temperature and heating time). Prior studies have investigated the kinetics of thermal injury of hepatocellular carcinoma, prostate cancer and renal carcinoma cells at temperatures in the range of 37–63 °C using different heating modalities [21,36,37,38]. However, there have been few reports of the kinetics of thermal injury to pancreatic cancer cells.

Overall, we have shown that exposure to heat stress decreased cell viability in pancreatic cancer cells (i.e., KPC and Pan02), in agreement with other in vitro and in vivo studies examining hyperthermia’s effectiveness as a potential therapeutic modality for treating pancreatic cancer [39,40,41,42,43,44,45]. As expected, the rate of decline in cell viability was more rapid as the applied temperature increased. KPC cells exhibited slightly greater resistance to thermal stress than the STO cells, indicated by their higher cell viabilities following heat treatment, while Pan02 cells showed the most resistance to heat treatment. We quantified the cell viability at 6 h and 24 h post heat exposure to visualize the progression of heat-induced cell death over time. For all three cell lines, the viability continued to decrease dramatically at 24 h post exposure for high temperature exposures (i.e., T = 50 °C, t > 5 min, T = 46 °C, t > 20 min) compared to the viability at 6 h post exposure. Baumann and colleagues [46] also exposed pancreatic cancer cells (PANC-1 and BxPC-3) to 45–50 °C for 5 min and measured the viability in different time points up to 7 days post exposure. The results were similar to ours, showing that near-complete cell death can occur following exposure to high temperatures (e.g., 50 °C), where complete cell death was not observed immediately post treatment but instead took longer to fully manifest in vitro. Ludwig et al. [44] also assessed the effect of hyperthermia on BxPC-3 human pancreatic cancer cells and showed that exposure to hyperthermia treatment at 41 °C and 43 °C for 1 h have almost no impact on cell viability, which was also reflected in our measured in vitro results.

Lage et al. [47] investigated the thermal sensitivity of human gastric (EPG85-257) and pancreatic carcinoma (EPP85-181) cell lines using water bath hyperthermia and calculated the Arrhenius injury model parameters. However, in their study, hyperthermia temperature was limited to 45 °C. In the present study, the optimized values for activation energy (*E_a_*) and frequency factor (A) in murine pancreatic cancer cells (i.e., KPC and Pan02) were calculated under near-isothermal heating conditions. The obtained kinetic coefficients were aligned with the Wright’s line plot of the Arrhenius coefficients from Pearce [48]. The range of coefficient of determination in this study (0.95 < *R*^2^ < 0.98, see Figure 4) for temperatures between 42.5–50 °C is similar to the values derived from other hyperthermia studies (0.95 < *R*^2^ for T > 40 °C) [16,49], indicating the suitability of the Arrhenius model for predicting thermally-induced injury in pancreatic cancer cell lines in vitro.

Similar to O’Neill [25] and Feng [19], we observed the initial shoulder region where cell viability was not affected at low temperatures with short durations, hence an improved Arrhenius model was used to provide a better fit, since traditional Arrhenius parameters (activation energy and frequency factor) calculated from low temperature, long duration exposures may not accurately predict cell death resulting from high temperature, low duration exposures [16,38]. Calculated RMSE values for all three cell lines were considerably improved when switching to the improved Arrhenius model from the traditional Arrhenius model. The thermal dose was also calculated using *R_CEM_* that was derived from our temperature-dependent cell survival data. The calculated *R_CEM_* values for KPC and Pan02 cells were 0.588 and 0.596, respectively. This was in agreement with the results presented by Mouratidis et al. [50], where the *R_CEM_* value for human colon cancer cell lines were calculated to be in the range of 0.5–0.53 at temperatures above 43 °C.

We also assessed the suitability of the two-state injury model by Feng et al. [19] for predicting changes in viability following heating of pancreatic cancer cells. The results presented by Feng reasonably accurately demonstrate the shoulder region of cell viability curves in their study on PC3 cell lines. However, in our study a rather poor fit between the two-state model and our collected in vitro data was observed, as illustrated in Figure 5. This might be due to the limited number of temperatures considered in this study, as the model relies on additional measured data at longer heating times where the viability tends to drop dramatically. Moreover, Feng et al. point out that the Arrhenius fit might actually provide a better estimation of cell viability at the higher temperatures where the shoulder region is not relevant. As previously described by Pearce [18], inclusion of more thermodynamic states may improve the accuracy of the two-state model.

Our study was limited to monolayer cell cultures, which may not accurately represent tumor cell response to heating in vivo. Previous in vivo studies have demonstrated a lower thermal threshold for the destruction of tumors when compared to cell culture in vitro under thermal exposure profiles [21,51,52]. The thermal damage model coefficients reported in this manuscript may be used to guide the selection of time-temperature profiles that can be anticipated to yield a specified level of thermal damage in pancreatic tumors, although caution should be taken when applying results to the clinical scenario, given the use of murine cell lines in the present study. Notably, this study highlighted variable susceptibility of different cell lines to hyperthermic exposure. Pancreatic tumors exhibit relatively high inter-tumor and intra-tumor heterogeneity [53]; understanding the differential thermal susceptibility of various cell populations to thermal exposure can inform prediction of the range of thermal damage levels anticipated for a given time-temperature profile delivered in the clinical scenario. Furthermore, in the clinical setting, thermal profiles are likely to vary across the targeted tumor due to the constraints of practical heating technology. Given time-temperature profiles observed during heating that can be measured with MRI or other thermometry techniques, quantitative analysis of thermal damage profiles can be performed using the reported thermal damage coefficients. Such analyses, coupled with post-treatment imaging of the targeted tumors, can provide means to assess and interpret treatment response [54].

## 5. Conclusions

We measured the extent of thermal injury in murine pancreatic cancer cell lines after exposure to temperatures in the range of 42.5–50 °C as informed by in vitro studies and derived thermal injury kinetic model parameters. Our results suggest that the improved Arrhenius model incorporating the time delay [24] to address the shoulder region is most suitable for use in mild hyperthermia therapies up to 60 min of heating. Finally, the accuracy of our developed injury predictive models was experimentally validated when cells were subjected to time-temperature profiles similar to those anticipated at the periphery of an ablation zone.

## Figures and Tables

**Figure 1 cancers-15-00655-f001:**
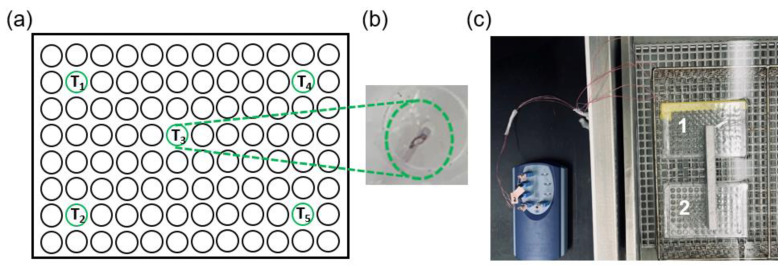
(**a**) Dummy plate design with five thermocouples for monitoring temperature during hyperthermia sealed within four corner wells and one central well. (**b**) Photograph of a thermocouple sealed within a well. (**c**) Cell-containing plate (plate 2) and dummy plate (plate 1) immersed within the water bath during hyperthermia.

**Figure 2 cancers-15-00655-f002:**
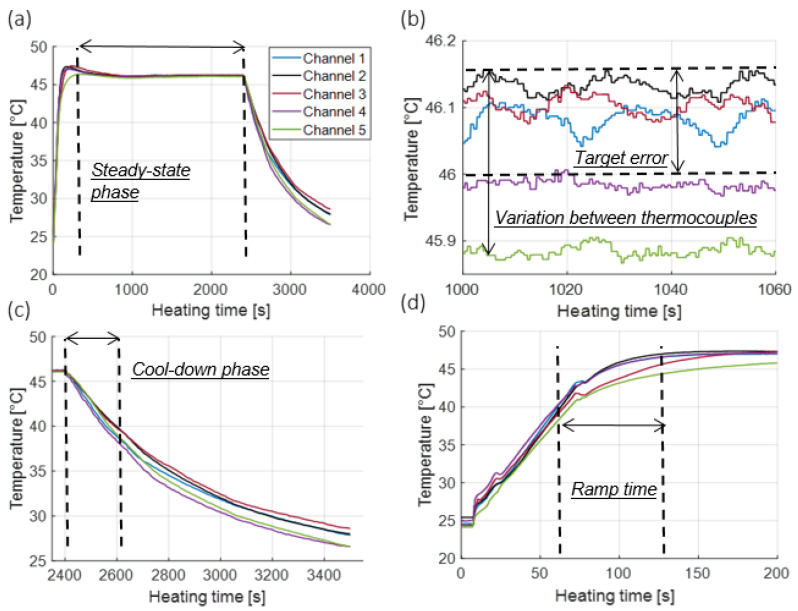
(**a**) Illustration of temperature recorded by thermocouples in the dummy plate during a 46 °C, 40 min hyperthermia exposure (**b**) illustration of temperatures over 1 min of the steady-state phase (**c**) illustration of temperatures during the cool-down phase, and (**d**) illustration of temperatures during the heat-up phase.

**Figure 3 cancers-15-00655-f003:**
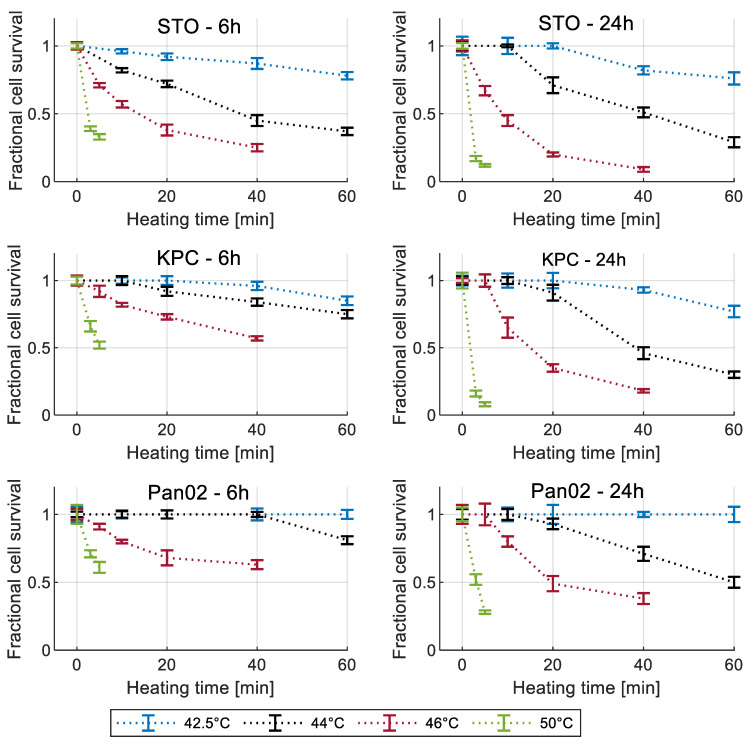
Measured cell viability for all three cell lines normalized to 37 °C control for different recovery times. Error bars represent one standard deviation.

**Figure 4 cancers-15-00655-f004:**
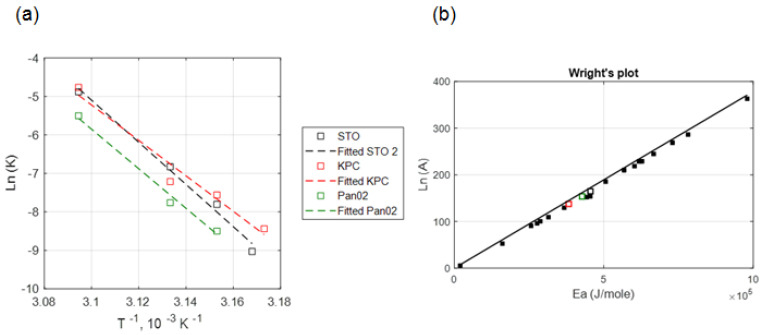
(**a**) Correlation between the kinetic coefficients, ln (A) and *E_a_* (KPC + Pan02 + STO), (**b**) comparison between the Wright’s plot (relationship between Arrhenius coefficients based on the literature) shown in solid black square markers and obtained Arrhenius coefficients for three different cell lines in our study. Hollow black, red and green square markers indicate the obtained Arrhenius coefficients for STO, KPC and Pan02 cells, respectively at 24 h post treatment in our study.

**Figure 5 cancers-15-00655-f005:**
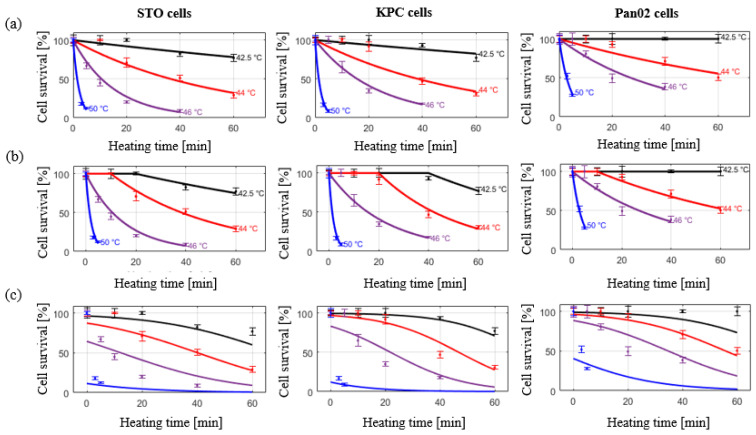
Cell viability assessed at 24 h post in vitro hyperthermia exposure in KPC, Pan02, and STO cell lines. Markers indicate measured data points. (**a**) Solid lines represent the simple Arrhenius model; (**b**), solid lines represent the improved Arrhenius model with time delay; (**c**) solid lines represent the predictive two- state model.

**Figure 6 cancers-15-00655-f006:**
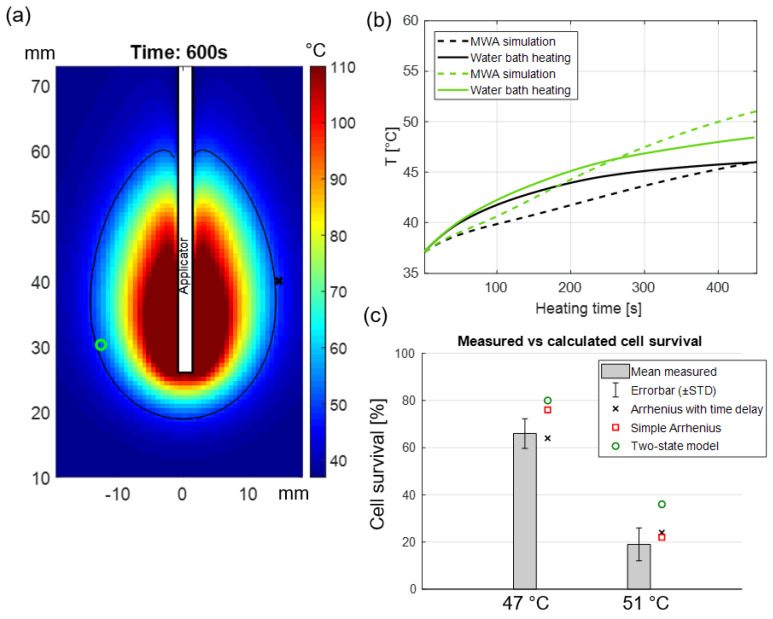
(**a**) Simulated temperature map in perfused pancreas tissue following 40 W, 10 min MWA, the black contour indicates the regions where 50 °C was achieved while the green circle and red x illustrate two positions along the periphery of the ablation zone where time-temperature history over 10 min was analyzed; (**b**) temperature profiles calculated from the bio-heat transfer model (shown in dashed lines) during 10 min MWA as well as experimentally measured temperatures in vitro during water bath hyperthermia (solid lines); (**c**) comparison between measured and calculated cell survival.

**Table 1 cancers-15-00655-t001:** Assessment of transient temperature profiles during in vitro heating.

Target Temperature (°C)	Mean Error (°C) (Min–Max)	Mean Variation between Thermocouples (°C) (Min–Max)	Mean Ramp Time (s) (Min–Max)	Mean Cool-Down Phase (s) (Min–Max]
42.5	0.2 (0.1–0.3)	0.2 (0.05–0.35)	85 (65–130)	130 (95–165)
44	0.25 (0.1–0.45)	0.35 (0.05–0.55)	75 (55–110)	320 (300–340)
46	0.25 (0.15–0.35)	0.15 (0.1–0.3)	75 (50–150)	267 (230–300)
50	0.18 (0.1–0.3)	0.2 (0.15–0.45)	80 (65–125)	320 (290–380)

**Table 2 cancers-15-00655-t002:** Obtained Arrhenius coefficients for all three cell lines.

Cell Type	*E_a_* (J/Mole)	*A* (s^−1^)	*b*	*m*
STO	455,630	e^164.79^	127,460	400
KPC	383,112	e^137.63^	254,920	800
Pan02	427,712	e^153.63^	127,460	400

**Table 3 cancers-15-00655-t003:** Calculated *R_CEM_* values for pancreatic cancer cells compared to other cell lines.

Cell Types in Our Study	R (T > 43 °C)	Cell Type from the Literature	R (T > 43 °C)
STO (Mice fibroblasts)	0.607	Prostate tumor cells [21]	0.474–50.624
KPC (Mice pancreatic tumor)	0.588	Baby hamster kidney cells [34]	0.550
Pan02 (Mice pancreatic tumor)	0.596	Porcine kidney cells [35]	0.596

## Data Availability

The data presented in this study are available on request from the corresponding author.

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
