# Peer review of "In Vitro Measurement and Mathematical Modeling of Thermally-Induced Injury in Pancreatic Cancer Cells"

_cancers, 2023, doi:10.3390/cancers15030655_

Round 1

Reviewer 1 Report

In this paper, the authors investigated the thermal injury parameters for pancreatic cancer cell lines in vitro. On this basis, the suitability of these parameters for predicting changes in cell viability after heating is investigated. They found that the improved Arrhenius model with time delay is the most suitable model to predict the viability of living cells in mild hyperthermia therapies. This is very meaningful for theoretical studies in the field of thermal therapy to evaluate the performance of different models or methods. The paper is well-written and organized. I suggest the paper to be accepted after minor revision. My only concern is that the results of the simulated temparture of a whole liver (Figure 6) was presented without detailed information. It is suggested to add more information about the heat transfer model, parameters, and the numerical method.

Reviewer 2 Report

I would advise the authors to make some assumptions about the clinical use of their observations to avoid that these interesting experiences are understood only as laboratory experiments. Pls add some hypothesis in the discussion about clinical strategy .

In other words, can the authors define better devices and methodologies to be applied in the clinic to treat patients with pancreatic cancer?

What about non isothermal heating to the heterogeneity of the malignancy itself ?
